# Association of Genetic Polymorphisms in *TLR3*, *TLR4*, *TLR7*, and *TLR8* with the Clinical Forms of Dengue in Patients from Veracruz, Mexico

**DOI:** 10.3390/v12111230

**Published:** 2020-10-29

**Authors:** Araceli Posadas-Mondragón, José Leopoldo Aguilar-Faisal, Gerardo Zuñiga, Jonathan Javier Magaña, José Angel Santiago-Cruz, Edith Guillén-Salomón, Verónica Alcántara-Farfán, María Luisa Arellano-Flores, Juan Santiago Salas-Benito, Rocío M. Neri-Bazán, Lucero Luna-Rojas, Amanda Marineth Avila-Trejo, Adolfo Chávez-Negrete

**Affiliations:** 1Laboratorio de Medicina de Conservación de la Sección de Estudios de Posgrado e Investigación, Escuela Superior Medicina, Instituto Politécnico Nacional, Plan de San Luis, Colonia Casco de Santo Tomas, Delegación Miguel Hidalgo, Ciudad de México 11340, Mexico; aposadasm1301@alumno.ipn.mx (A.P.-M.); rocionerib@gmail.com (R.M.N.-B.); lunar.32@hotmail.com (L.L.-R.); ammarat.gdds@gmail.com (A.M.A.-T.); 2Departamento de Zoología, Escuela Nacional de Ciencias Biológicas, Instituto Politécnico Nacional, Plan de Ayala, Colonia Casco de Santo Tomas, Miguel Hidalgo, Ciudad de México 11340, Mexico; capotezu@hotmail.com; 3Laboratorio de Medicina Genómica, Departamento de Genética, Instituto Nacional de Rehabilitación-LGII, Calzada México Xochimilco No. 289, Colonia Arenal de Guadalupe, Ciudad de México 14389, Mexico; maganasm@hotmail.com; 4Departamento de Microbiología, Escuela Nacional de Ciencias Biológicas, Instituto Politécnico Nacional, Plan de Ayala, Colonia Casco de Santo Tomas, Delegación Miguel Hidalgo, Ciudad de Mexico 11340, Mexico; jasc0210@hotmail.com; 5Coordinación de Planeación y Enlace Institucional, Delegación Veracruz Norte, Instituto Mexicano del Seguro Social, Lomas del Estadio S/N Xalapa, Veracruz 91090, Mexico; edith.guillen@imss.gob.mx; 6Departamento de Bioquímica, Escuela Nacional de Ciencias Biológicas, Instituto Politécnico Nacional, Plan de Ayala, Colonia Casco de Santo Tomas, Delegación Miguel Hidalgo, Ciudad de México 11340, Mexico; veroalf@yahoo.com; 7Unidad de Investigación en Enfermedades Endocrinas, Hospital de Especialidades, Centro Médico, Nacional siglo XXI, Instituto Mexicano del Seguro Social, Cuauhtémoc 330, Colonia Doctores, Delegación Cuauhtémoc, Ciudad de México 06720, Mexico; maluif9@yahoo.com; 8Laboratorio de Biomedicina Molecular III (Virología) de la Sección de Estudios de Posgrado e Investigación, Escuela Nacional de Medicina y Homeopatía, Instituto Politécnico Nacional, Guillermo Massieu Helguera 239, La Escalera, Gustavo A. Madero, Ciudad de México 07320, Mexico; jsalasb@yahoo.com; 9Coordinación de Educación Médica Continua, Comité Normativo Nacional de Medicina General, Cuauhtémoc 330, Colonia Doctores, Delegación Cuauhtémoc CDMX 06720, Mexico; achavezn@gmail.com

**Keywords:** toll-like receptor, dengue, gene polymorphisms, dengue fever, dengue hemorrhagic fever, human genetic susceptibility

## Abstract

Dengue manifestations range from a mild form, dengue fever (DF), to more severe forms such as dengue hemorrhagic fever (DHF) and dengue shock syndrome (DSS). The ability of the host to present one of these clinical forms could be related to polymorphisms located in genes of the Toll-like receptors (TLRs) which activate the pro-inflammatory response. Therefore, the genotyping of single nucleotide genetic polymorphisms (SNPs) in *TLR3* (rs3775291 and rs6552950), *TLR4* (rs2737190, rs10759932, rs4986790, rs4986791, rs11536865, and rs10983755), *TLR7* (rs179008 and rs3853839), and *TLR8* (rs3764880, rs5741883, rs4830805, and rs1548731) was carried out in non-genetically related DHF patients, DF patients, and general population (GP) subjects. The SNPs were analyzed by real-time PCR by genotyping assays from Applied Biosystems^®^. The codominance model showed that dengue patients had a lower probability of presenting the *TLR4*-rs2737190-G/G genotype (odds ratio (OR) (95% CI) = 0.34 (0.14–0.8), *p* = 0.038). Dengue patients showed a lower probability of presenting *TLR4*-rs11536865-G/C genotype (OR (95% CI) = 0.19 (0.05–0.73), *p* = 0.0092) and had a high probability of presenting the TACG haplotype, but lower probability of presenting the TGCG haplotype in the *TLR4* compared to GP individuals (OR (95% CI) = 0.55 (0.35–0.86), *p* = 0.0084). In conclusion, the *TLR4*-rs2737190-G/G and *TLR4*-rs11536865-G/C genotypes and TGCG haplotype were associated with protection from dengue.

## 1. Introduction

Dengue fever is a disease caused by the dengue virus, a positive sense single-stranded RNA virus, which has four serotypes, DENV-1, DENV-2, DENV-3, and DENV-4 [1]. The disease can present as dengue fever (DF), which is characterized by fever, headache, and retro-ocular, muscular and joint pain, in addition to nausea, vomiting, and a rash. Some patients may develop dengue hemorrhagic fever (DHF), for which the clinical picture is potentially fatal; symptoms include plasma extravasation, fluid accumulation, respiratory distress, and severe hemorrhages or organ failures.

Dengue hemorrhagic fever has been associated with infections by the serotype DENV-2 due to its genetic variation, replication efficiency, and genotype. In addition, the disease may be aggravated by the presence of pre-existing heterotypic antibodies; when the virus-antibody complexes bind to the Fc receptors, they facilitate the infection of target cells, causing an increase in viraemia and cytokine levels [2]. It has been documented that single nucleotide genetic polymorphisms (SNPs) in cytokines (*TNF*-308, *TGFβ1*-509) and receptors (*DC*-*SIGN*-336) are associated with susceptibility to DHF [3,4,5].

The host innate immune response is the main defense against viral infection, and is activated by Toll-like receptors (TLRs). These are type I transmembrane glycoproteins, with a domain rich in repeats of leucine and a domain homologous to the interleukin 1 receptor [6]. TLRs activate the response factor of interferon and NF-kB, resulting in the transcription of type I interferon and proinflammatory cytokines [7,8]. In particular, TLR3, TLR7, and TLR8 are expressed in endosomal compartments when viral oligonucleotides are present [8]. TLR3 responds to double-stranded RNA, a product of viral replication [9,10], whereas TLR7 and TLR8 recognize non-methylated single-stranded viral RNA, rich in guanosine or uridine (ssRNA) [11,12]. In contrast, TLR4 is an extracellular receptor that is activated by the NS1 protein of the dengue virus [13].

SNPs in the *TLRs* affect the immune response because they can modify the structure of the protein or the expression of the gene, giving rise to a wide variety of clinical forms of infectious diseases, including viral diseases [14,15,16]. The SNPs present in the *TLRs* associated with dengue disease are poorly documented. A study conducted of children in Indonesia showed that the presence of the Asp299Gly and Thr399Ile SNPs in *TLR4* favors the susceptibility to infection and the severity of dengue in this population [17]. A study conducted with adults in India showed that individuals heterozygous with Asp299Gly and Thr399Ile had greater susceptibility to dengue infection [18]. Likewise, the high frequency of the T allele of the SNP *TLR3*-rs3775291 in adults of India associated with DHF in comparison with DF has been reported. By comparison, the *TLR8* C-A haplotype of the SNPs rs3764879 and rs3764880 was found to be associated with DF in adult males from India [19].

In this work, we studied the association of SNPs in *TLRs* with the clinical forms of dengue in the adult Mexican population. We included those SNPs that have been found to be associated with dengue in other populations: *TLR3* (located in chromosome 4): rs3775291 (exonic, T/C, L412F); *TLR4* (located in chromosome 9): rs4986791 (exonic, C/T, T399I), and rs4986790 (exonic, A/G, D299G); and *TLR8* (located in chromosome X): rs3764880 (exonic, A/G, M1V); in addition to those SNPs that have been associated with other infectious diseases: *TLR3*: rs6552950 (intronic, A/G); *TLR4*: 10,759,932 (promoter −1607, C/T), rs2737190 (promoter −2570, A/G), rs11536865 (promoter −728, C/G), and rs10983755 (promoter −2081, A/G); *TLR7* (located in chromosome X): rs179008 (exonic, A/T, Q11L), and rs3853839 (UTR-3, C/G); and *TLR8*: rs5741883 (intronic, C/T), rs4830805 (intronic, A/G), and rs1548731 (intronic, C/T). The analysis showed that *TLR4*-rs2737190-G/G and *TLR4*-rs11536865-G/C genotypes and TGCG haplotype were associated with protection from dengue.

## 2. Materials and Methods

### 2.1. Study Design

Blood samples were collected during the epidemic outbreaks of dengue between August and October 2013 and September to October 2015, in the Family Medicine Unit No. 61 and the General Hospital No. 71 of the Mexican Institute of Social Security of Veracruz, Mexico. The study included unrelated volunteers, residents of Veracruz, over 18 years of age. The participants were classified as having DHF (*n* = 65) or DF (*n* = 100), these being defined according to the criteria described by the WHO, 1997 [20], namely, DF—platelet count > 100 × 10^9^/L and DHF—platelet count <100 × 10^9^/L, plus any hemorrhagic manifestation. The clinical diagnosis was confirmed by serology and identification of the viral genome. Blood samples were obtained during the acute phase of the disease (1–8 days after the onset of symptoms). Subjects from the general population (GP, *n* = 89) were also included in the study. The samples of general population subjects correspond to healthy adults (over 18 years of age) who attended a routine medical check-up in the health units mentioned above, and showed normal levels in the hematic biometry test. Samples of general population subjects and dengue patients were taken in the same health units at the same time.

This study was approved by the National Commission of Scientific Research of the Mexican Social Security Institute (Registration No. 2010-785-041) and by the Local Committee of Research and Ethics in Health Research Unit No. 3003 of the Family Medicine Unit No. 61 of Veracruz (Registration No. 2810-009-018). The procedures followed were in accordance with the ethical standards of the Helsinki Declaration of the World Medical Association. The patient’s written consent was obtained and all information was handled anonymously.

### 2.2. Blood Samples

Blood samples were collected in Vacutainer^®^ tubes with and without an anticoagulant for the obtained serum and leukocytes, respectively. The serum and leukocytes were separated from tubes by centrifugation at 1.157× *g* for 15 min in an LMC-3000 Biosan centrifuge (Biosan, Riga, Latvia).

### 2.3. Hematological Values

Platelets, leukocytes, neutrophils, and monocytes were determined using the COULTER LH 500 Hematology Analyzer (Beckman Coulter, Indianapolis, IN, USA).

### 2.4. Serotype-Specific Detection of Dengue Viruses

Viral RNA was extracted from the participants’ sera and assayed with a QIAamp Viral RNA mini kit (QIAGEN Cat. No. 52906, Germantown, MD, USA). Subsequently, the viral genome for the four serotypes was identified by multiplex quantitative RT-qPCR. Primers comprising flavivirus and four fluorogenic TaqMan probes specific for each serotype of DENV in the NS5 region were used [21]. A superscript III Platinum enzyme One-Step qRT-PCR Kit (Cat. No. 1732088; Invitrogen, Thermofisher, Austin, TX, USA) was used. The analyses were performed in a LightCycler 480 instrument (Roche, Indianapolis, IN, USA). The strains DENV-1 (Hawaii), DENV-2 (New Guinea), DENV-3 (H-87), and DENV-4 (H-241) were used as positive controls.

### 2.5. Anti-Dengue IgM and IgG Capture ELISA

IgM and IgG anti-dengue in the participants’ sera were determined using capture ELISA kits EIA-3470 and EIA-3471 (DRG Diagnostics, Springfield, NJ, USA).

### 2.6. Anti-Dengue IgG Subclasses EIA

DENV-2 (New Guinea) was used to infect Vero cells. The infected cells were cultured in Dulbecco’s Modified Eagle’s medium supplemented with antibiotics and 1% FBS until a cytopathic effect was evident. Subsequently, the cells were harvested and centrifuged for 15 min at 4500× *g*, 4 °C, to remove cells and cell debris, and the resultant virus-rich supernatant was stored in PBS in aliquots at −70 °C until use in an IgG subclass capture enzyme immunoassay. The protein concentration of the DENV-2 antigen was determined with the Bradford assay as 10 mg/mL. An internal capture EIA system was developed to determine the anti-dengue IgG subclasses. The antigen was diluted with a carbonate buffer solution (NaHCO_3_ and Na_2_CO_3_ 1 M, pH 9.6) to a concentration of 10 µg/mL. A 96-well polyvinyl microplate was coated for 24 h at 4 °C with the Vero cell-derived virus (see above). The microwells were then washed three times with a washing buffer (0.5% (*vol/vol*) Tween 20 in PBS). The plates were blocked with 5% (*wt/vol*) skim milk in PBS for 1 h at 37 °C and serum samples diluted 1:100 in PBS supplemented with 2% (*wt/vol*) skim milk. The microwells were then washed again three times. One hundred microliters of diluted serum samples was then added to the microwells and incubated for 1 h at 37 °C. The microwells were then washed again three times. Antihuman mAbs IgG1, IgG2, IgG3, and IgG4 conjugated with HRP (ab99774, ab99779, ab99829, and ab99817; Abcam, San Francisco, CA, USA) diluted 1:2000 in PBS-skim milk were added to the wells for 1 h at 37 °C. The microwells were then washed again three times before they were developed with o-phenylenediamine dihydrochloride as the substrate. The reaction was stopped by adding 2 N H_2_SO_4_ (100 µL/well) and the absorbance read at 492 nm.

### 2.7. Genotyping of Polymorphisms in TLR3, TLR4, TLR7, and TLR8

Genomic DNA was extracted from the leukocytes using the Gentra Puregene Blood Kit (Catalog No. 158389, QIAGEN^®^). The SNPs were analyzed by real-time PCR using Applied Biosystems^®^ genotyping assays. Genotyping was done for the SNPs: *TLR4*-rs2737190, *TLR4*-rs10759932, *TLR4*-rs4986790, *TLR4*-rs4986791, *TLR4*-rs11536865, *TLR4*-rs10983755; *TLR3*-rs3775291, *TLR3*-rs6552950; *TLR7*-rs179008, *TLR7*-rs3853839; *TLR8*-rs3764880, *TLR8*-rs5741883, *TLR8*-rs4830805, and *TLR8*-rs1548731.

### 2.8. Calculations and Statistical Analysis

Anti-dengue IgM and IgG positive and negative samples were designated according to a cut-off control, which was provided by the manufacturer. Analyses were performed in duplicate and the absorbance averaged to obtain the cut-off value. Samples were considered positive when the absorbance was higher than 10% of the cut-off value and negative when it was lower than 10% of the cut-off value.

Detection of IgG subclasses was performed only in samples which anti-dengue IgG antibodies were detected. The analysis was performed in duplicate, blank and conjugate controls were included in the study design. The indices of the anti-dengue IgG subclasses were calculated using the following formula: Index = (OD of sample − OD of blank)/(OD of blank), where OD is optical density, in such a way that the values greater than zero were taken as positive.

The anti-dengue IgM, IgG, and IgG subclasses indices were categorized as high or low using the tertile method. IgM/IgG ratios were also calculated to enable classification as primary or secondary infection according to the WHO criteria. Dengue infection was defined as primary if the IgM/IgG OD ratio was greater than 1.2 (sera diluted 1:100) and secondary if the ratio was less than 1.2 [22].

Values are expressed as the mean ± SD for each study group. Statistical significance was assessed by the Mann–Whitney test and the odds ratio (OR) was used to determine associations. GraphPad Prism version 5.0 program (GraphPad Software, La Jolla, CA, USA) was used for statistical analysis. *p*-values < 0.05 were considered to denote significance.

Allelic and genotypic frequencies, Hardy Weinberg equilibrium, association of the SNPs with the disease, interaction between the SNPs and the covariates, linkage disequilibrium, haplotype frequencies, and association of the SNPs and haplotypes with the disease were performed with the server SNP stats. The risk of every genotype was analyzed by five models: Codominant, dominant, recessive, over-dominant, and additive; to choose the inheritance model that best fits the data we followed the Akaike information criterion (AIC). Logistic regression models were used for the analysis of interactions between the SNPs and the covariates. The two-stage iterative method, expectation maximization (EM) algorithm, was used to estimate the haplotype frequencies. The association between haplotypes and disease were analyzed via logistic regression. *TLR7* and *TLR8* are located on the X chromosome so the analysis was done separately for men and for women.

The study of the sites of transcription factor binding in the DNA sequence where the SNPs *TLR4*-rs2737190 and *TLR4*-rs11536865 are located was performed using the virtual laboratory PROMO [23,24].

## 3. Results

### 3.1. Hematological Values of DHF and DF Patients

Hematological findings in patients with DHF and DF are shown in Table 1. By definition, patients with DHF had thrombocytopenia. They also presented leukopenia (OR with 95% CI = 2.118 (1.096–4.09), *p* < 0.001), and neutropenia (OR with 95% CI = 2.947 (1.281–6.779), *p* < 0.0001). However, DHF patients had a significantly higher percentage of lymphocytes (OR with 95% CI = 3.467 (1.244–9.663) *p* < 0.0001) and monocytes (OR with 95% CI = 2.095 (1.075–4.083), *p* = 0.0001).

### 3.2. Identification of Viral Serotypes in Patients with DF and DHF

The viral genome was identified in 54 samples (33%) from 165 patients in the acute phase. Of 46 samples from patients with DF, three (6.5%) had DENV-1, 42 (91.3%) DENV-2, and one (2.2%) DENV-4, whereas eight samples from patients with DHF were identified as having DENV-2.

### 3.3. IgM, Total IgG, and IgG Subclasses in Patients with DF and DHF

Anti-dengue IgM was detected in 37% (37/100) of the sera of the patients with clinical diagnoses of DF and in 63% (41/65) of those with DHF. Anti-dengue IgG was detected in 93% (93/100) of the sera of patients with clinical diagnoses of DF and in 96.9 (63/65) of those with DHF.

In total, findings in 27% (27/100) and 29.23% (19/65) of the sera of DF and DHF patients, respectively, were classified as indicating primary infection, whereas 73% (73/100) and 70.77% (46/65) of the sera of patients with DF and DHF, respectively were classified as indicating secondary infection. The IgM/IgG ratio mean was 5.71 for primary infection and 0.34 for secondary infection.

Tertile analysis showed that the cut-offs for IgM, total IgG, IgG1, IgG2, IgG3, and IgG4 were 1.47, 1.678, 0.564, 0.018, 0.033, 0.193, respectively. Immunoglobulin indices higher than the cut-off points were considered high and indices lower than the cut-off point were considered low. Interestingly, DHF patients had high levels of IgM (58.46%, *n* = 38) in addition to IgG (55.38%, *n* = 36), and a high percentage of patients with DF were negative for IgM (74%, *n* = 74) and had low levels of IgG (77%, *n* = 77). A high percentage of patients with DF and DHF had low IgG1 levels, 71.8% and 61.02%, respectively. However, a greater number of patients with DHF had high levels of IgG1 with respect to DF, 38.98% and 28.2%, respectively. Most of the patients with DF and DHF were negative for IgG2, and only 8.97% and 15.25% were positive with high levels, respectively. Most DHF patients had low IgG3 levels (38.98%, *n* = 23), whereas DF patients had high levels (38.46%, *n* = 30). Both DF and DHF patients had low IgG4 levels, 69.23% and 59.32%, respectively. However, there was a higher percentage of DHF patients with high IgG4 levels (35.59%, *n* = 21) (Table 2).

### 3.4. Association of Genetic Polymorphisms in TLR3, TLR7, and TLR8 with the Clinical Forms of Dengue Virus Infection

In *TLR3*, the ancestral alleles *TLR3*-rs3775291-C and *TLR3*-rs6552950-A appeared with greater frequency than the minor alleles *TLR3*-rs3775291-T and *TLR3*-rs6552950-G. The most frequent genotypes were *TLR3*-rs3775291-C/C, *TLR3*-rs3775291-C/T, *TLR3*-rs6552950-A/A, and *TLR3*-rs6552950-A/G (Appendix A).

*TLR7* analysis showed that the ancestral allele *TLR7*-rs179008-A and the minor alleles *TLR7*-rs3853839-G, *TLR7*-rs179008-A/A, *TLR7*-rs179008-A/T, and *TLR7*-rs3853839-G/C genotypes were the most frequent in the groups (Appendix A).

Compared to *TLR8*-rs3764880, the ancestral and minor allele *TLR8*-rs3764880 G and the allele A were presented with almost the same frequency in the groups. The *TLR8*-rs3764880-G/G and *TLR8*-rs3764880-G/A genotypes were the most frequent in the groups. The ancestral alleles *TLR8*-rs5741883-C, *TLR8*-rs4830805-G, and *TLR8*-rs1548731-C appeared more frequently in the groups. The most frequent genotypes were *TLR8*-rs5741883-C/C, *TLR8*-rs4830805 G/A, *TLR8*-rs4830805 G/G, and *TLR8*-rs1548731-C/C (Appendix A).

The allelic and genotypic frequencies followed the Hardy Weinberg equilibrium, with the exception of *TLR7*-rs3853839 for the dengue (DEN) group. There was no statistical association in the crude analysis with clinical forms of the dengue disease.

### 3.5. Association of Genetic Polymorphisms in the TLR4 Gene with the Clinical Forms of Dengue Virus Infection

*TLR4* SNP analysis showed that the ancestral alleles of *TLR4*-rs4986790, *TLR4*-rs4986791, *TLR4*-rs10759932, and *TLR4*-rs10983755, in addition to the *TLR4*-rs4986790-A/A, *TLR4*-rs4986791-C/C, *TLR4*-rs10759932-T/T, and *TLR4*-rs10983755-G/G genotypes, were the most frequent in the groups (Appendix A). No statistical association was found with the clinical forms of the disease and the SNPs mentioned. The allelic and genotypic frequencies followed the Hardy Weinberg equilibrium, with the exception of *TLR4*-rs4986790 and *TLR4*-rs10983755.

Allelic analysis of *TLR4*-rs11536865 showed that allele G was more frequent than allele C. DF, DHF, and DEN patients had a higher probability of presenting allele G than GP individuals (OR (95% CI) = 4.65 (0.975–22.25), *p* = 0.034; 6.07 (0.75–49.1), *p* = 0.055; OR (95% CI) = 5.12 (1.34–19.5), *p* = 0.0081, respectively). The most frequent genotype was G/G. DEN group patients showed lower probability of presenting the G/C genotype (OR (95% CI) = 0.19 (0.05–0.73), *p* = 0.0092) than GP individuals (Table 3).

According to the analysis of *TLR4*-rs2737190, allele A was more frequent in the studied groups. The most frequent genotypes were A/A and A/G. In the crude analysis of association with the clinical forms of the disease, it was found that in a codominant and recessive model, when comparing the groups GP vs. DF, and GP vs. DEN, the G/G genotype was associated with protection from DF and DEN (Table 3).

The codominance model showed that DF patients had a lower probability of presenting the G/G genotype, OR (95% CI) = 0.29 (0.10–0.84), *p* = 0.035 and an AIC value of 260.6, compared to GP individuals. This was also observed in a recessive model, OR (95% CI) = 0.29 (0.11–0.78), *p* = 0.0095, and an AIC value of 258.6 (Table 4).

The codominance model showed that DEN patients had a lower probability of presenting the G/G genotype, OR (95% CI) = 0.34 (0.14–0.8), *p* = 0.038, and an AIC value of 328.5, compared to GP individuals. This was also observed in a recessive model, OR (95% CI) = 0.36 (0.16–0.8), *p* = 0.011, and an AIC value of 326.6 (Table 5).

### 3.6. Linkage Disequilibrium Analysis for TLR4 SNPs

The SNPs rs10759932-rs2737190, rs2737190-rs4986791, rs10759932-rs11536865, and rs2737190-rs11536865 showed a positive correlation according to the Pearson correlation coefficient (r ˃ 0); in addition, there was significant difference in the χ^2^ test. However, there was a negative correlation (r ˂ 0) between the SNPs rs10759932-rs4986791 and rs4986791-rs11536865 but no statistically significant difference was found with the χ^2^ test (Table 6).

### 3.7. Haplotype Analyses in TLR4 SNPs

The haplotype analysis in the *TLR4* generated eight haplotypes; the most frequent in the four groups are shown in Table 7. The individuals with DF, DHF, and DEN groups had a high probability of presenting the TACG haplotype in the *TLR4* with respect to GP individuals; and had lower probability of presenting the TGCG haplotype in the *TLR4* compared to GP individuals (OR (95% CI), 0.55 (0.34–0.91), *p* = 0.021; 0.59 (0.34–1.0), *p* = 0.05; 0.55 (0.35–0.86), *p* = 0.0084; respectively) (Table 8).

### 3.8. Interaction Analysis of TLR SNPs and the Hematological Covariates

We performed an interaction analysis of *TLR* SNPs and the levels of hematological covariates leukocytes, lymphocytes, monocytes, and immunoglobulin levels of IgM, total IgG, IgG1, IgG2, IgG3, IgG4, and primary and secondary dengue infections.

#### 3.8.1. *TLR3*

DF and DEN patients with low levels of leukocytes and lymphocytes were more likely to be *TLR3*-rs3775291-C/C and *TLR3*-rs6552950-A/A genotypes than those of GP. In addition, DF, DHF, and DEN patients with high monocyte levels had a higher probability of being *TLR3*-rs3775291-C/C and *TLR3*-rs6552950-A/A genotypes than those of GP (Figure 1).

DHF patients with high IgM were more likely to be *TLR3*-rs6552950-A/A and *TLR3*-rs6552950-A/G genotypes. DHF patients with high IgG were more likely to be *TLR3*-rs3775291-C/C, *TLR3*-rs3775291-C/T, *TLR3*-rs6552950-A/A, and *TLR3*-rs6552950-A/G genotypes. DHF patients with high IgG1 levels were more likely to be *TLR3*-rs3775291-C/T and *TLR3*-rs6552950-A/G genotypes. On the other hand, DHF patients who presented a secondary infection had a lower probability of being *TLR3*-rs3775291-T/T and *TLR3*-rs6552950-A/G genotypes compared to DF patients (Figure 1).

#### 3.8.2. *TLR4*

DHF patients with low levels of leukocytes had a higher probability of being *TLR4*-rs10759932-T/T, *TLR4*-11536865-G/G, *TLR4*-rs2737190-A/A, and *TLR4*-rs4986791-C/C genotypes. In addition, DF and DEN patients with high levels of monocytes were more likely to be *TLR4*-rs10759932-T/T, *TLR4*-11536865-G/G, *TLR4*-rs2737190-A/A, *TLR4*-rs2737190-A/G, and *TLR4*-rs4986791-C/C genotypes than GP individuals. Furthermore, DEN patients with low levels of lymphocytes were more likely to be *TLR4*-rs10759932-C/T, *TLR4*-11536865-G/G, *TLR4*-rs2737190-A/G, and *TLR4*-rs4986791-C/C genotypes than GP individuals. DHF patients with high levels of IgM had a higher probability of being *TLR4*-11536865-G/G and *TLR4*-rs4986791-C/C genotypes. DHF patients with high levels of IgG were more likely to be *TLR4*-11536865-G/G, *TLR4*-rs2737190-A/A, and *TLR4*-rs4986791-C/C genotypes. DHF patients with high levels of IgG1 had a higher probability of being *TLR4*-rs2737190-G/G genotype than DF patients. DHF patients with secondary infections had a lower probability of being *TLR4*-rs10759932-T/T, *TLR4*-11536865-G/G, *TLR4*-rs2737190-A/G, and *TLR4*-rs4986791-C/C genotypes than those with DF (Figure 2).

#### 3.8.3. *TLR7*

DHF women patients with low levels of leukocytes were more likely to be *TLR7*-rs179008-A/T and *TLR7*-rs3853839-C/C. DHF women patients with high levels of monocytes were more likely to be *TLR7*-rs179008-A/A and *TLR7*-rs3853839-C/C. DHF women patients with high levels of IgG were more likely of being *TLR7*-rs179008-A/A, *TLR7*-rs179008-A/T, *TLR7*-rs3853839-C/G, *TLR7*-rs3853839-C/C genotypes; however, those with high levels of IgG3 had a lower probability of presenting *TLR7*-rs179008-A/A and *TLR7*-rs179008-A/T genotypes. DHF patients with high levels of IgG1 were more likely to be *TLR7*-rs3853839-C/C. In addition, DF and DEN patients with high levels of monocytes had a high probability of being the *TLR7*-rs179008-A/A genotype with respect to GP individuals (Figure 3).

#### 3.8.4. *TLR8*

DHF women patients with a high monocyte and high IgG levels had a greater probability of being *TLR8*-rs3764880-A/A. Moreover, DF and DEN women patients with a high monocyte and low lymphocyte levels were more likely to be *TLR8*-rs3764880-A/G; DEN, *TLR8*-rs5741883-C/C, *TLR8*-rs4830805-A/A, and *TLR8*-rs1548731-C/C genotypes than GP. DHF women patients with low levels of leukocytes were more likely to be *TLR8*-rs5741883 C/C, *TLR8*-rs4830805-G/G, *TLR8*-rs4830805-A/G, *TLR8*-rs1548731-C/C, and *TLR8*-rs1548731-C/T genotypes. DHF women patients with low levels of monocytes and high IgG were more likely to be *TLR8*-rs5741883 C/C and *TLR8*-rs1548731-C/C genotypes. DHF patients with secondary infections had a lower probability of being *TLR8*-rs1548731-C/C genotype with respect to DF. Moreover, those with high IgG3 levels had a lower probability of presenting *TLR8*-rs5741883 C/C, *TLR8*-rs1548731-C/C, *TLR8*-rs1548731-C/T, and *TLR8*-rs1548731-C/C with respect to DF (Figure 4).

### 3.9. Study of the Binding of Transcription Factors to the Promoter Region of TLR4-rs2737190 and TLR4-rs11536865

Study of binding transcription factors showed that when the alleles *TLR4*-rs2737190-G or *TLR4*-rs2737190-A are present, the transcription factors GR-beta, HNF-1C, FOXP3, HNF-1B, Pax-5, and P53 join to the site; however, when the allele *TLR4*-rs2737190-A is present, which was most frequent in DF and DHF, in addition to the previous transcription factors, others are added: HOX D9, HOX D10, PR-B, and PR-A (Figure 5). Regarding the *TLR4*-rs11536865 SNP, when the allele C is present, only the glucocorticoid receptor transcription factor (GR) joins to the site, but when the allele G is present, the transcription factors GR and GR-α join to the site (Figure 6).

## 4. Discussion

Several factors have been described that may be associated with the development of the clinical forms of dengue, particularly those due to the virus, the environment, and the host. Regarding the host, the most relevant factors are the immunological status and polymorphisms in genes related to the immune response [4]. In a previous article, we analyzed the immunological status of dengue patients and found that IgG1 increased significantly in the sera of DHF patients with secondary infections. Furthermore, IgG4 was significantly higher in the sera of DHF patients who were in the early and late acute phase of both primary and secondary infection [25]. The relationship between innate and adaptive immunity is of interest, therefore, in the current study, we address the role of genetic polymorphisms in Toll-like receptors (TLRs) and their interaction with hematological variables and immunoglobulins.

*TLR3*-rs3775291 (T/C) implies the change of leucine by phenylalanine (L412F); interestingly we found that the C (ancestral) allele occurred more frequently than the allele T (minor) in the population. In addition, the most frequent genotypes in the four studied groups were C/C and C/T; the T/T genotype was found in a very low percentage in DHF patients. Dengue patients with low levels of leukocytes and lymphocytes, and high levels of monocytes and IgG, were more likely to be *TLR3*-rs3775291-C/C. However, DHF patients with high IgG1 levels were more likely to be *TLR3*-rs3775291-C/T, and who presented a secondary infection had a lower probability of being *TLR3*-rs3775291-T/T. In the Indian population, the T allele has been associated with a lower risk of DHF in a dominant mode [19]. It has also been associated with a protective effect against HIV infection among highly exposed HIV seronegative subjects and intravenous drug users [26]. By contrast, the C/C genotype has been associated with severe recurrence of hepatitis C after liver transplantation [27], and the C/T genotype was reported more frequently in children with human cytomegalovirus infection [28]. It is likely that this SNP may be related to a lower risk of developing severe dengue.

*TLR3*-rs6552950 involves the G/A transition and is located in the intron region. We found that the allele A (ancestral) was the most frequent in the population and, in particular, in the DHF group (76%). A high percentage of DHF patients were A/A genotype. Dengue patients with low levels of leukocytes and lymphocytes, and high levels of monocytes, IgM, and IgG, were more likely to be *TLR3*-rs6552950-A/A. However, DHF patients with high IgG1 levels were more likely to be *TLR3*-rs6552950-A/G, and who presented a secondary infection had a lower probability of being *TLR3*-rs6552950-A/G. Interestingly, allele G has been associated with severe chikungunya fever compared to patients with mild chikungunya fever, and neutralization assays of specific antibodies against chikungunya virus revealed that patients with the G/G genotype had significantly less neutralizing antibodies compared to patients with genotypes A/A or A/G [29]. It would be interesting to perform an in vitro study of the neutralizing or enhancing capacity of the antibodies of DHF patients with the A/A genotype.

The *TLR7*-rs3853839-G/C genotype and G allele were the most frequent in the study groups. DHF women patients with low levels of leukocytes, and high levels of total IgG and IgG1, were more likely to be *TLR7*-rs3853839-C/C. In the same way, the *TLR7*-rs3853839-C/C genotype has been demonstrated to have a significant association with dengue infectivity in Indian patients [30]. A similar pattern was observed in this study as that in the study of chikungunya virus infected Indian patients, in which the G/C genotype was significantly higher [31].

*TLR8*-rs3764880 implies the transition A/G and the change of methionine by valine. We found that the A/G genotype was the most frequent in women with DHF. DHF women patients with high monocyte and high IgG levels had a greater probability of being *TLR8*-rs3764880-A/A and rs3764880-A/G, respectively. In the Indian population, a greater frequency of the *TLR8* haplotype *TLR8*-rs3764879-rs3764880 (C-A) was observed in males with DF than in healthy men [19]. In addition, the AA/A genotype has been previously associated with the development of warning signs among dengue infected patients in the Indian population [30]. The role of TLR4 in response to viruses has not been widely explored; it is known that the surface glycoprotein of respiratory syncytial virus (RSV), the glycoprotein of the Ebola virus, and the envelope protein of the mouse mammary tumor virus (MMTV) can activate TLR4 [32,33,34,35]. In addition, the hexameric form of NS1 protein of dengue virus activates mouse macrophages and human peripheral blood mononuclear cells (PBMCs) via TLR4 with the consequent release of proinflammatory cytokines [13]. TLR4 signaling seems to play an important role in the balance between the ability to control infections and in the pathogenesis of inflammatory diseases.

The *TLR4*-rs4986790 (+896A/G) and *TLR4*-rs4986791 (+1196C/T) have been described that change the ligand-binding site of the TLR4. *TLR4*-rs4986790 shows an amino acid substitution of Gly in the place of Asp at the 299 position, and *TLR4*-rs4986791 the change of Ile for Thr at the 399 position. In this study, the *TLR4*-rs4986790-A/A and *TLR4*-rs4986791-C/C genotypes were the most frequent in the groups. Dengue patients with high levels of monocytes and low levels of lymphocytes, and DHF patients with high levels of IgM and IgG and low levels of leukocytes, had a higher probability of being *TLR4*-rs4986791-C/C. However, DHF patients with secondary infections had a lower probability of being rs4986791-C/C. Compared to *TLR4*-rs4986790, the allelic and genotypic frequencies did not follow the Hardy Weinberg equilibrium. However, previous research in the Indian population shows that individuals with the heterozygous genotype for *TLR4*-rs4986790 and *TLR4*-rs4986791 polymorphisms had increased susceptibility to dengue infection and the IIe/Gly (TG) haplotype was associated with the risk of the disease [18]. However, a study from Indonesia showed that the distribution of the *TLR4*-rs4986790 and *TLR4*-rs4986791 polymorphisms did not differ between children with severe dengue case presentation (DHF/DSS) and controls [17].

The *TLR4*-rs11536865 polymorphism (promoter-728, C/G) had not been associated with infectious diseases; however, a systematic analysis of this study shows that dengue patients had a higher probability of presenting allele G than GP individuals. The most frequent genotype was G/G, and dengue patients showed lower probability of presenting the G/C genotype. In addition, dengue patients with high levels of monocytes and low levels of lymphocytes, and DHF patients with high levels of IgM and IgG, were more likely to be *TLR4*-11536865-G/G. However, DHF patients with secondary infections had a lower probability of being *TLR4*-11536865-G/G. According to these results, the *TLR4*-rs11536865-G allele was the most frequent in a study of African American liver donors, in which this polymorphism exhibited a significant association with liver graft failure [36]. Analysis of the transcription factors that bind to the site where the *TLR4*-rs11536865 polymorphism (promoter-728, C/G) is located showed that when the allele C is present only the glucocorticoid receptor transcription factor (GR) joins to the site, but when the allele G is present, the transcription factors GR and GR-α join to the site. The anti-inflammatory effects of steroids are mediated through the glucocorticoid receptor-α isoform (GR-α), which suppresses expression of inflammatory genes through mechanisms known as transactivation or transrepression. Transactivation occurs through direct binding of activated GR-α to DNA sequences called glucocorticoid responsive elements (GREs). Transrepression defines a direct interaction of activated GR-α with different transcription factors, such as NF-κB and AP-1, repressing their abilities to induce expression of proinflammatory genes. It would be interesting to understand the role of GR-α in the pathogenesis of DHF [37].

*TLR4*-rs2737190 (G/A transition) is located in the promoter region of *TLR4*. The allele A was the most frequent in the population studied. The *TLR4*-rs2737190-A/A was the most frequent in DHF, and the *TLR4*-rs2737190-A/G in DF. Interestingly, the *TLR4*-rs2737190-G/G genotype was frequent in the general population but was less frequent in individuals with DF and DHF. Dengue patients with high levels of monocytes and low levels of lymphocytes were more likely to be *TLR4*-rs2737190-A/A and A/G. DHF patients with low levels of leukocytes and high levels of total IgG had a higher probability of being *TLR4*-rs2737190-A/A. However, DHF patients with high levels of IgG1 had a higher probability of being *TLR4*-rs2737190-G/G. DHF patients with secondary infections had a lower probability of being *TLR4*-rs2737190-A/G. It has been reported that people with *TLR4*-rs2737190-A/G genotype had a higher risk of clinical tuberculosis or smear-positive tuberculosis [38]. However, it has been shown that preterm infants carrying the allele *TLR4*-rs2737190-G are prone to infections by Gram-negative bacteria [39].

In this study, it was found that in a recessive model individuals with DEN and, in particular, DF are less likely to be a *TLR4*-rs2737190-G/G genotype. This indicates that two copies of the variant allele are necessary to change the risk. Therefore, the heterozygous and homozygous genotypes of the most frequent allele have the same effect. According to the analysis of the transcription factors that bind to the site where this polymorphism is located, it was observed that in the presence of the allele *TLR4*-rs2737190-G, the transcription factor HNF-1B binds; however, when the allele A is present, five more transcription factors bind to the site: HOX D9, HOX D10, progesterone receptors PR-B, and PR-A. PR-A and PR-B progesterone receptors have been associated with the clinical outcome of viral diseases such as hepatitis E, HIV, and influenza. In women infected with HIV-1, contraceptive pills containing progesterone increase the susceptibility to the infection in vivo. It has also been observed that the treatment with progesterone increases the replication of HIV-1 in cells [40]. The reduced expression of PR and a high viral load result in a precarious pregnancy in hepatitis E [41]. In the influenza infection, progesterone measured the production of epidermal growth factor anfiregulin by epithelial cells and the repair of lung tissue after infection [42]. It would be of interest to understand the role of these transcription factors in the pathogenesis of DHF.

The combination of SNPs could have a severe impact on the immune system. To know if the SNPs studied in each chromosome are linked and can form haplotypes, the analysis of linkage disequilibrium was undertaken. The haplotype analysis in the *TLR4* showed that the individuals within the DF, DHF, and DEN groups had a high probability of presenting the TACG haplotype in *TLR4* with respect to the GP individuals, and had a lower probability of presenting the TGCG haplotype in *TLR4* compared to the GP individuals.

## Figures and Tables

**Figure 1 viruses-12-01230-f001:**
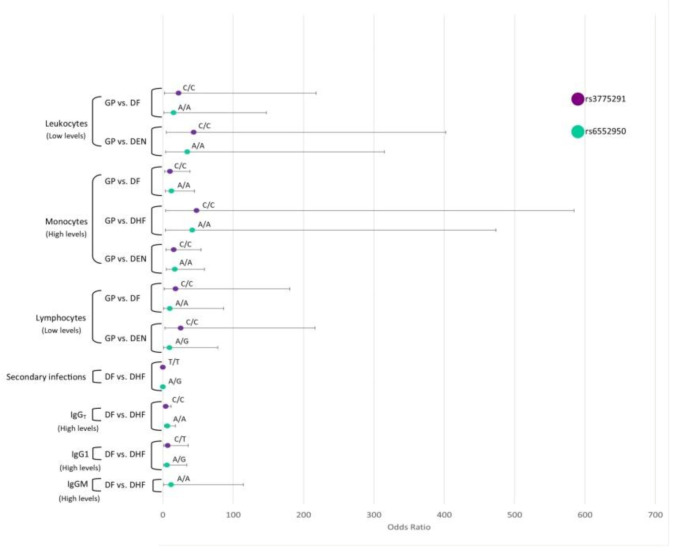
Interaction analysis of *TLR*3 SNPs and the hematological covariates. Interaction analysis of *TLR3* SNPs and the levels of hematological covariates leukocytes, lymphocytes, monocytes, and immunoglobulin levels of IgM, total IgG, IgG1, and secondary dengue infections. Odds ratios and confidence intervals are shown in abscissa axis and genotypes are indicated above the odds ratios point.

**Figure 2 viruses-12-01230-f002:**
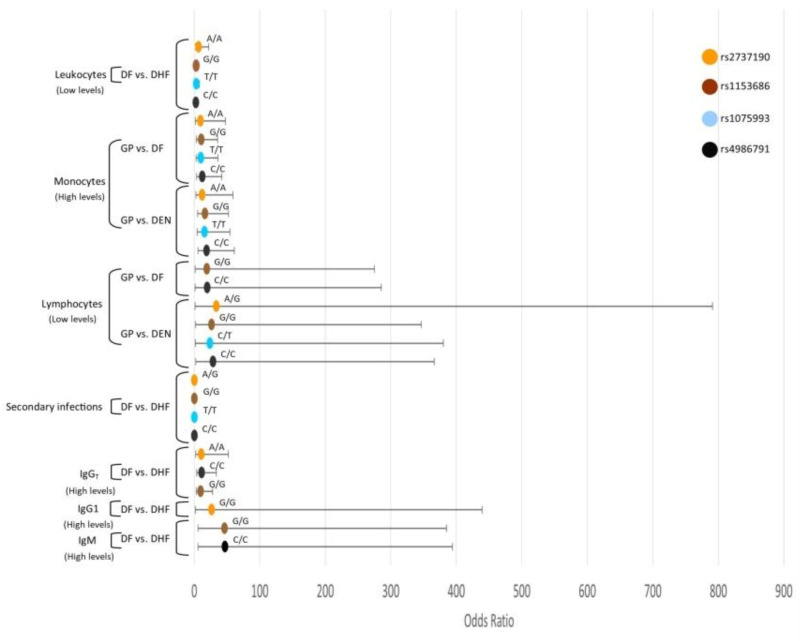
Interaction analysis of *TLR*4 SNPs and the hematological covariates. Interaction analysis of *TLR4* SNPs and the levels of hematological covariates leukocytes, lymphocytes, monocytes, and immunoglobulin levels of IgM, total IgG, IgG1, and secondary dengue infections. Odds ratios and confidence intervals are shown in abscissa axis and genotypes are indicated above the odds ratios point.

**Figure 3 viruses-12-01230-f003:**
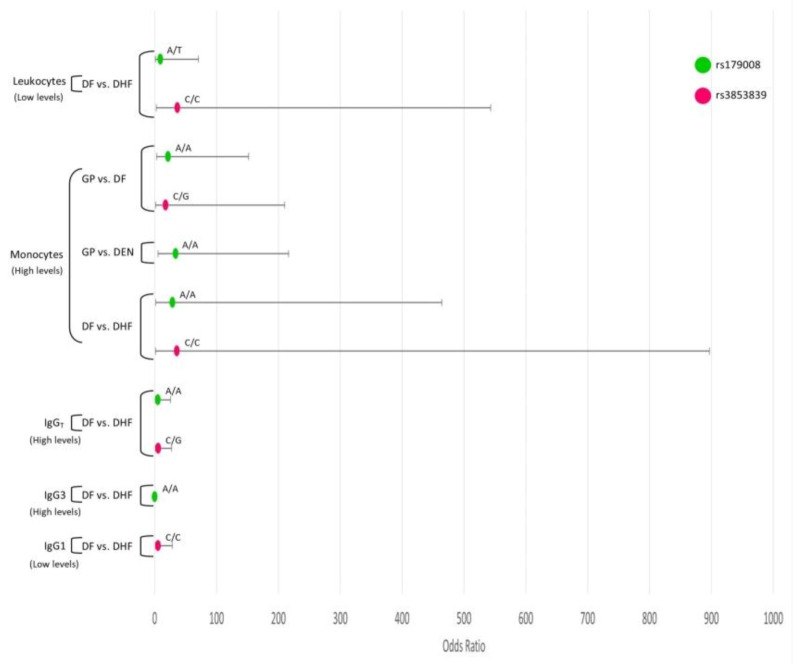
Interaction analysis of *TLR*7 SNPs and the hematological covariates. Interaction analysis of *TLR7* SNPs and the levels of hematological covariates leukocytes, lymphocytes, monocytes, and immunoglobulin levels of total IgG, IgG1, IgG3. Odds ratios and confidence intervals are shown in abscissa axis and genotypes are indicated above the odds ratios point.

**Figure 4 viruses-12-01230-f004:**
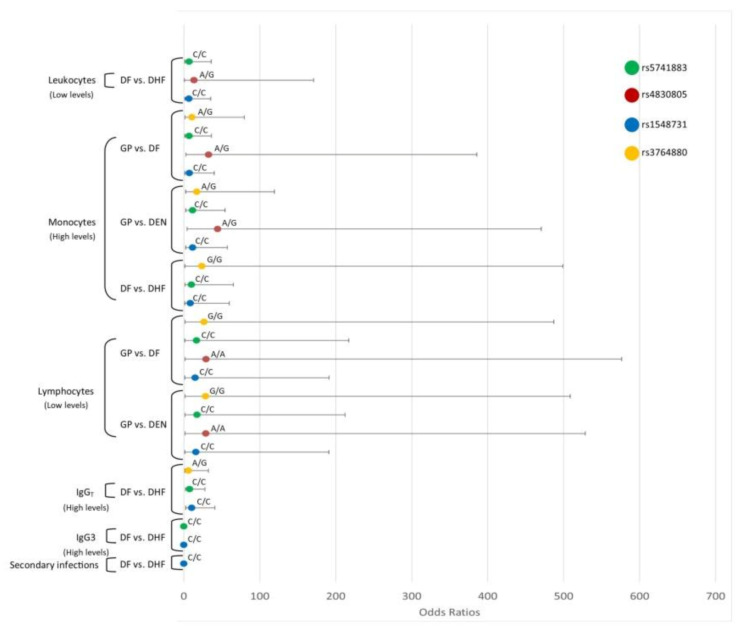
Interaction analysis of *TLR*8 SNPs and the hematological covariates. Interaction analysis of *TLR8* SNPs and the levels of hematological covariates leukocytes, lymphocytes, monocytes, and immunoglobulin levels total IgG, IgG3. Odds ratios and confidence intervals are shown in abscissa axis and genotypes are indicated above the odds ratios point.

**Figure 5 viruses-12-01230-f005:**
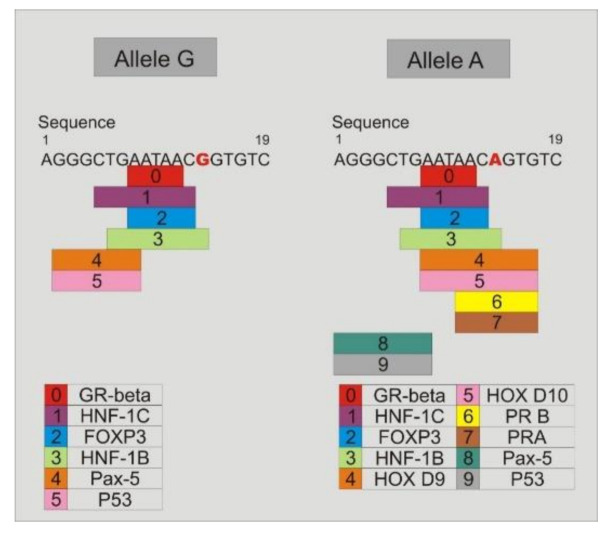
Study of the transcription factor binding sites in the DNA sequence where the *TLR4*-rs2737190 is located. The transcription factor HNF-1B binds in the presence of the allele *TLR4*-rs2737190-G. When the allele A is present, five more transcription factors bind to the site: HOX D9, HOX D10, and progesterone receptors PR-B and PR-A.

**Figure 6 viruses-12-01230-f006:**
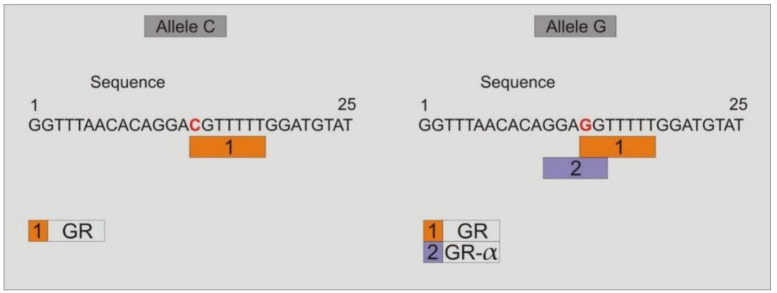
Study of the transcription factor binding sites in the DNA sequence where the *TLR4*-rs11536865 is located. When the allele C is present only the glucocorticoid receptor transcription factor (GR) joins to the site, but when the allele G is present, the transcription factors GR and GR-α join to the site.

**Table 1 viruses-12-01230-t001:** Hematological values of the individuals with dengue fever (DF) and dengue hemorrhagic fever (DHF), and general population (GP).

	DF (*n* = 100)	DHF (*n* = 65)	Dengue (*n* = 165)	GP (*n* = 89)
Age	37.86 ± 13.49 ^a,b^	42.95 ± 14.50 ^a^	39.85 ± 14.07 ^c^	49.52 ± 22.99 ^b,c^
Female/male	49/51	28/37	77/88	59/30
Platelets (×10^9^/L)	191.1 ± 65.07 ^a,b^	54.03 ± 30.48 ^a,c^	137.1 ±86.21 ^d^	253.7 ± 64.34 ^b,c,d^
Leukocytes (×10^9^/L)	6.279 ± 3.618 ^a,b^	4.459 ± 1.964 ^a,c^	5.562 ± 3.194 ^d^	7.926 ± 2.050b ^c,d^
Neutrophils (%)	61.79 ± 16.84 ^a,b^	46.49 ± 16.62 ^a,c^	55.76 ± 18.31	56.76 ± 10.51 ^b,c^
Lymphocytes (%)	24.77 ± 14.38 ^a,b^	35.23 ± 14.88 ^a^	28.89 ± 15.41	31.24 ± 9.044 ^b^
Monocytes (%)	10.33 ± 4.705 ^a,b^	14.44 ± 7.11 ^a, c^	11.95 ± 6.095 ^d^	7.517 ± 2.646 ^b,c,d^

GP: General Population; DF: Dengue Fever; DHF: Dengue Hemorrhagic Fever; DEN: Dengue. The statistical differences (*p* ˂ 0.0001) between the groups are indicated by pairs of the same letters.

**Table 2 viruses-12-01230-t002:** Classification of antibody levels according to Tertile analysis.

Antibody	Clinical Dengue Forms	Indices		
		**Low *n* (%)**	**High *n* (%)**	**Negative *n* (%)**
IgM	DF	12 (12)	14 (14)	74 (74)
	DHF	4 (6.15)	38 (58.46)	23 (35.38)
IgG	DF	77 (77)	16 (16)	7 (7)
	DHF	27 (41.53)	36 (55.38)	2 (3.077)
IgG1	DF	56 (71.8)	22 (28.2)	0
	DHF	36 (61.02)	23 (38.98)	0
IgG2	DF	0	7 (8.97)	71 (91.03)
	DHF	0	9 (15.25)	50 (84.75)
IgG3	DF	21 (26.92)	30 (38.46)	27 (34.62)
	DHF	23 (38.98)	15 (25.42)	21 (35.59)
IgG4	DF	54 (69.23)	24 (30.77)	0
	DHF	35 (59.32)	21 (35.59)	3 (5.08)

DF: Dengue Fever; DHF: Dengue Hemorrhagic Fever.

**Table 3 viruses-12-01230-t003:** Allelic and genotypic frequencies of *TLR4*-rs2737190 and *TLR4*-rs11536865.

*TLR4* SNP Genotype/Allele	GP (*n* = 89) N (%)	DF (*n* = 100) N (%)	DHF (*n* = 65) N (%)	DEN (DF + DHF) (*n* = 165) N (%)	DF vs. DHF	GP vs. DF	GP vs. DHF	GP vs. DEN
rs2737190								
G ^Mi,An^	72 (40)	64 (32)	40 (31)	104 (32)	0.94(0.58−1.52)	0.69 (0.45−1.05)	0.65 (0.40−1.05)	0.67 (0.46–0.98)
A	106 (60)	136 (68)	90 (69)	226 (68)				
*p*-value					0.81	0.0875	0.08	0.043
A/A	33 (37)	42 (42)	31 (48)	73 (44)	1.0	1.0	1.0	1.0
A/G	40 (45)	52 (52)	28 (43)	80 (48)	0.73 (0.38−1.40)	1.02 (0.55−1.89)	0.75 (0.37−1.48)	0.9 (0.52−1.58)
G/G	16 (18)	6 (6)	6 (9)	12 (8)	1.35 (0.40−4.60)	0.29 (0.10−0.84)	0.40 (0.14−1.15)	0.34 (0.14−0.8)
*p*-value					0.47	0.035	0.21	0.038
HWE (*p*-value)	0.52	0.067	1	0.15				
rs11536865								
G ^An^	170 (96)	198 (99)	129 (99)	327 (99)	1.3 (0.12−14.53)	4.65 (0.975−22.25)	6.07 (0.75–49.1)	5.12 (1.34–19.5)
C ^Mi^	8 (4)	2 (1)	1 (1)	3 (1)				
*p*-value					0.83	0.034	0.055	0.0081
G/C	8 (9)	2 (2)	1 (2)	3 (2)	0.77 (0.07−8.62)	0.21 (0.04−1.0)	0.16 (0.02−1.30)	0.19 (0.05−0.73)
G/G	81 (91)	98 (98)	64 (98)	162 (98)	1.0	1.0	1.0	1.0
*p*-value					0.83	0.028	0.035	0.0092
HWE (*p*-value)	1	1	1	1				

GP: General Population; DF: Dengue Fever; DHF: Dengue Hemorrhagic Fever; DEN: Dengue; W: Women; M: Men; Mi: Minor Allele; An: Ancestral Allele; HWE: Hardy Weinberg Equilibrium.

**Table 4 viruses-12-01230-t004:** Association of *TLR4*-rs2737190 with the response group (DF) (*n* = 189).

Model	Genotype	GP	DF	OR (95% IC)	*p*-Value	AIC
Codominant	A/A	33 (37.1%)	42 (42%)	1.0	0.035	260.6
	A/G	40 (44.9%)	52 (52%)	1.02 (0.55–1.89)		
	G/G	16 (18%)	6 (6%)	0.29 (0.10–0.84)		
Recessive	A/A-A/G	73 (82%)	94 (94%)	1.00	0.0095	258.6
	G/G	16 (18%)	6 (6%)	0.29 (0.11–0.78)		

GP: General Population; DF: Dengue Fever; OR: Odds ratio; CI: Confidence Interval; AIC: Akaike information criterion.

**Table 5 viruses-12-01230-t005:** Association of TLR4-rs 2737190 with the response group (DEN) (*n* = 254).

Model	Genotype	GP	DEN	OR (95% IC)	*p*-Value	AIC
Codominant	A/A	33 (37.1%)	73 (44.2%)	1.0	0.038	328.5
	A/G	40 (44.9%)	80 (48.5%)	0.9 (0.52–1.58)		
	G/G	16 (18%)	12 (7.3%)	0.34 (0.14–0.8)		
Recessive	A/A-A/G	73 (82%)	153 (92.7%)	1.00	0.011	326.6
	G/G	16 (18%)	12 (7.3%)	0.36 (0.16–0.8)		

GP: General Population; DF: Dengue Fever; OR: Odds ratio; CI: Confidence Interval; AIC: Akaike information criterion.

**Table 6 viruses-12-01230-t006:** Linkage disequilibrium analysis for *TLR4* SNPs, GP vs. DEN.

	rs2737190	rs4986791	rs11536865
rs10759932	0.0642	−0.0013	0.0041
	0.999	0.842	0.212
	0.4533	−0.0352	0.0953
	104.38	0.629	4.61
	<2 × 10^−16^	0.4278	0.0318
	254	254	254
rs2737190		0.0102	0.006
		0.992	0.426
		0.1724	0.0870
		15.097	3.843
		0.0001	0.0499
		254	254
rs4986791	D		−0.0002
	D’		0.768
	r		−0.0145
	χ^2^		0.106
	*p*-Value		0.7446
	N		254

Linkage disequilibrium parameters D and D’ for each pair; D: Deviation between the expected haplotype frequency and the observed frequency; D’: D scaled in [−1, 1] range; r: Correlation coefficient between alleles.

**Table 7 viruses-12-01230-t007:** Estimation of the haplotype frequencies of the *TLR4* gene SNPs.

	rs10759932	rs2737190	rs4986791	rs11536865	GP (*n* = 89)	DF (*n* = 100)	DHF (*n* = 65)	DEN (*n* = 165)
1	T	A	C	G	0.56	0.68	0.6939	0.6848
2	T	G	C	G	0.2999	0.19	0.2	0.1939
3	C	G	C	G	0.084	0.1	0.0846	0.0939
4	T	G	T	G	0.0056	0.02	0.154	0.0182
5	T	A	C	C	0.0355	NA	NA	NA
6	T	G	C	C	0.0091	NA	NA	NA
7	C	G	C	C	3e-04	0.01	0.0077	0.0091
8	C	G	T	G	0.0056	NA	NA	NA

GP: General Population; DF: Dengue Fever; DHF: Dengue Hemorrhagic Fever; DEN: Dengue.

**Table 8 viruses-12-01230-t008:** Haplotype association of *TLR4* with the clinical forms of dengue.

Groups	rs10759932	rs2737190	rs4986791	rs11536865	OR (95% CI), *p*-Value	Global Haplotype Association *p*-Value
GP vs. DF	T	A	C	G	1.0	
	T	G	C	G	0.55 (0.34–0.91), *p*-Value 0.021	0.0095
GP vs. DHF	T	A	C	G	1.0	
	T	G	C	G	0.59 (0.34–1.0), *p*-Value 0.05	0.036
GP vs. DEN	T	A	C	G	1.0	
	T	G	C	G	0.55 (0.35–0.86), *p*-Value 0.0084	0.0019
DF vs. DHF	T	A	C	G	1.0	
	T	G	C	G	1.02 (0.57–1.82), *p*-Value 0.95	0.98

DF: Dengue Fever; DHF: Dengue Hemorrhagic Fever; DEN: Dengue; GP: General population; OR: Odds ratio; IC: Confidence interval.

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
