# Peer review of "Association of Genetic Polymorphisms in TLR3, TLR4, TLR7, and TLR8 with the Clinical Forms of Dengue in Patients from Veracruz, Mexico"

_viruses, 2020, doi:10.3390/v12111230_

Round 1
Reviewer 1 Report
This is an interesting manuscript describing the association of genetic polymorphisms in TLRs genes and dengue disease presentation. Overall, the manuscript needs to be improved by clarifying the methods and lightening the writing. Some aspects of the results section need a careful revision to avoid repetitive/ unnecessary information. Authors also need to better address the importance of looking into TLRs genes in dengue patients. Methods section needs more details, especially regarding assays used for dengue serology.
Title: Add country after Veracruz.
Lines 66-67: in primary dengue infections? Please clarify.
Lines 108-110: How were dengue cases defined? Based on laboratory confirmation or clinical/epidemiological evaluation?
Lines 110-111: Please add more information about the subjects from general population. Healthy adults? Where they were recruited? Matched by age/ ethnicity/ any other factor with dengue cases?
Lines 139-157: Please describe how in-house capture assay for IgG subclass detection was validated. Were positive and negative controls included? What was the cutoff for positivity?
Lines 196-205: Authors don’t need to repeat every information already shown in the table. It’s really hard to read through this paragraph with all the numbers (again, repeated in the table!).
Table 1: There is something missing. What are those superscript letters for?
Lines 210-212: Please indicate if there was variation in RNA positivity with time of sample collection.
Lines 221-223: Please clarify. If authors defined cutoff based on blank, anything lower than the cutoff should be negative, undetectable. A sample (or pool of samples) from a dengue naïve individual should be more appropriated to calculate cutoffs.
Table 2: IgG3 are known markers of recent dengue infection. How do authors justify the discrepancy between the results from IgM and IgG3 in the DF group?
Description of TLRs SNPs: Instead of repeating same information (same frequency between groups) I suggest focusing on what was different between groups and summarize the remaining.
Description of interaction analysis: Please consider adding ORs/95% CI to a supplementary table and summarize results. It’s not pleasant to read these paragraphs, too many numbers.
Discussion: Please consider focus on the main findings and what was different between groups.
Author Response
- Lines 66-67: in primary dengue infections? Please clarify
Dengue infection was defined as primary if the IgM/IgG OD ratio was greater than 1.2 (sera diluted 1:100) and secondary if the ratio was less than 1.2 [20]. (See lines 176-179). The definition of primary or secondary infection is important since hemorrhagic dengue cases have been mainly associated with secondary infections, however, in a previous study we found hemorrhagic dengue cases in primary infections, so we consider it interesting to associate this variable with SNPs are TLRs.
- Lines 108-110: How were dengue cases defined? Based on laboratory confirmation or clinical/epidemiological evaluation?
Dengue cases were defined based on laboratory confirmation and hematological values according to the criteria described by the WHO, 1997. (See lines 108-112)
This statement was added on lines 111 and 112: The clinical diagnosis was confirmed by serology and identification of the viral genome.
- Lines 110-111: Please add more information about the subjects from general population. Healthy adults? Where they were recruited? Matched by age/ ethnicity/ any other factor with dengue cases?
This statement was added on lines 114 and 117: The samples of general population subjects correspond to healthy adults (over 18 years of age) who attended a routine medical check-up in the health units mentioned above, and showed normal levels in the hematic biometry test. Samples of general population subjects and dengue patients were taken in the same health units at the same period.
- Lines 139-157: Please describe how in-house capture assay for IgG subclass detection was validated. Were positive and negative controls included? What was the cutoff for positivity?
This statement was added on lines 170-174: Detection of IgG subclasses was performed only in samples which anti-dengue IgG antibodies were detected. Analysis was performed in duplicate, blank and conjugate controls were included in the study design. The indices of the anti-dengue IgG subclasses were calculated using the following formula: index = (OD of sample − OD of blank)/(OD of blank), where OD is optical density, in such a way that the values ​​greater than zero were taken as positive. (Posadas M. A.; Aguilar F. J. L.; Chávez N. A.; Guillén S. E.; Alcántara F. V.; Luna R. L.; Ávila T. A. M.; Pacheco Y. J. C. Indices of anti-dengue immunoglobulin G subclasses in adult Mexican patients with febrile and hemorrhagic dengue in the acute phase. Microbiol Immunol. 2017;61(10):433–41.)
- Lines 196-205: Authors don’t need to repeat every information already shown in the table. It’s really hard to read through this paragraph with all the numbers (again, repeated in the table!).
The data that is repeated in the text and in the table have been eliminated from the paragraph.
- Table 1: There is something missing. What are those superscript letters for?
The following statement is indicated in the footer of the table: The statistical differences (P ˂0.0001) between the groups are indicated by the pairs of the same letters.
- Lines 210-212: Please indicate if there was variation in RNA positivity with time of sample collection.
The samples were kept frozen during transport and handled cold for processing, however, despite the fact that the patients were in the acute phase, it was not possible to identify the viral genome in all the samples by real-time RT-PCR.
- Lines 221-223: Please clarify. If authors defined cutoff based on blank, anything lower than the cutoff should be negative, undetectable. A sample (or pool of samples) from a dengue naïve individual should be more appropriated to calculate cutoffs.
Once the identification of positive samples IgM, total IgG, IgG1, IgG2, IgG3, and IgG4 was done, the tertile analysis was done to classify the data in high or low levels. Only positive samples were included for this analysis.
- Table 2: IgG3 are known markers of recent dengue infection. How do authors justify the discrepancy between the results from IgM and IgG3 in the DF group?
The presence of IgM antibodies is a marker of recent primary infection. However, IgG3 is a marker of recent infection but not necessarily a primary infection, this is important because the samples in this study were collected in the acute phase of the disease (before 8 days after symptoms started). Furthermore, it is important to consider that the samples were obtained from a dengue hyper-endemic region where a response would be expected, mainly IgG, as described in the results of this study, however, the presence of IgM in cases of DF and DHF is interesting for further study.
- Description of interaction analysis: Please consider adding ORs/95% CI to a supplementary table and summarize results. It’s not pleasant to read these paragraphs, too many numbers.
We have added forests plots of Odds Ratios and its confidence intervals (Figura 1-4)
Reviewer 2 Report
In the study design the last phrase “all information was anonymized as far as possible” should probably be reviewed.
The results are detailed, sometimes is not easy to follow because so much information is included, like OR and IC95% in text( for instance in point 3.8.1 TLR3, 3.8.2 TLR4).
The manuscript is relevant in terms of research in genetic polymorphisms in TLR in Dengue fever patients and the data will probably be used in clinical practice as markers of severity.
Author Response
- In the study design the last phrase “all information was anonymizedas far as possible” should probably be reviewed.
The phrase has been modified.
- The results are detailed, sometimes is not easy to follow because so much information is included, like OR and IC95% in text( forinstance in point 3.8.1 TLR3, 3.8.2 TLR4).
We have added forests plots of Odds Ratios and its confidence intervals (Figura 1-4)
Reviewer 3 Report
As a clinical infectious diseases Dr I found the information presented quite "data dense" and somewhat hard to follow - but I'm sure the details are relevant for other workers in the field. The overall message that polymorphisms of TLR4 are important in influencing the severity of dengue was well supported.
There were a few minor changes that I suggest.
Line 66 DHF can occur with any of the serotypes not just dengue 2. I can see that most of your cases with DHF had dengue 2 but I'm sure in other years that could be different.
Line 68 Heterotypic antibodies (not heteropic)
line 82 - The SNPs do not give rise to a wide variety of infectious diseases rather they influence the phenotype of those diseases.
Line 100- the sentence starting "The systematic analysis.." should have a reference
line 109 - reference 20 actually introduced a new classification for dengue and describe why the WHO had moved from the old classification that the authors have used. There are plenty of older references which might be more appropriate.
Line 167 - can you find a better term than "fabricant"?
Table 1 lymphocytes rather than "limphocytes"F
Author Response
- Line 66 DHF can occur with any of the serotypes not just dengue 2. Ican see that most of your cases with DHF had dengue 2 but I'm surein other years that could be different.
I agree with the idea that DHF outbreaks can be caused by the other serotypes, however, for this study the interaction of the different dengue serotypes with the SNPs was not analyzed because not in all subjects the serotype could be identified, so that in future research it would be important to make these associations.
The epidemiological surveillance system in Mexico reports that serotype 2 is the main cause of dengue hemorrhagic fever, including the area where the samples were obtained.
- Line 68 Heterotypic antibodies (not heteropic)
The word has been modified.
- Line 82 - The SNPs do not give rise to a wide variety of infectious diseases rather they influence the phenotype of those diseases.
The sentence has been modified.
- Line 100- the sentence starting "The systematic analysis.." should have a reference
The statement that begins with "The systematic analysis ..." refers to the results that were obtained in this research, so no reference was added, however the word “systematic” has been removed to avoid confusion.
- Line 109 - References 20 actually introduced a new classification for dengue and describe why the WHO had moved from the old classification that the authors have used. There are plenty of older references which might be more appropriate.
We have added the following reference: World Health Organization. (1997). Dengue haemorrhagic fever : diagnosis, treatment, prevention and control, 2nd ed. World Health Organization.
- Line 167 - can you find a better term than "fabricant"?
The term has been modified.
- Table 1 lymphocytes rather than "limphocytes"F
The word has been modified
Round 2
Reviewer 1 Report
Authors have addresses previous comments. This reviewer is satisfied with the responses.